# Surface association sensitizes *Pseudomonas aeruginosa* to quorum sensing

Sara K. Chuang[1], Geoffrey D. Vrla[2], Kathrin S. Fröhlich[2,3] & Zemer Gitai [2]

In the pathogen *Pseudomonas aeruginosa*, LasR is a quorum sensing (QS) master regulator that senses the concentration of secreted autoinducers as a proxy for bacterial cell density. Counterintuitively, previous studies showed that saturating amounts of the LasR ligand, 3OC12-HSL, fail to induce the full LasR regulon in low-density liquid cultures. Here we demonstrate that surface association, which is necessary for many of the same group behaviors as QS, promotes stronger QS responses. We show that *lasR* is upregulated upon surface association, and that surface-associated bacteria induce LasR targets more strongly in response to autoinducer than planktonic cultures. This increased sensitivity may be due to surface-dependent *lasR* induction initiating a positive feedback loop through the small RNA, Lrs1. The increased sensitivity of surface-associated cells to QS is affected by the type IV pilus (TFP) retraction motors and the minor pilins. The coupling of physical surface responses and chemical QS responses could enable these bacteria to trigger community behaviors more robustly when they are more beneficial.

[1] Department of Chemical and Biological Engineering, Princeton University, Princeton, NJ 08540, USA. [2] Department of Molecular Biology, Princeton University, Princeton, NJ 08540, USA. [3] Department of Biology I, Microbiology, Ludwig-Maximilians-University Munich, D-82152 Martinsried, Germany. Correspondence and requests for materials should be addressed to Z.G. (email: zgitai@princeton.edu)

  1

Bacteria grow in a complex world and are constantly exposed to diverse stimuli. For many years the field has characterized the response of bacteria to the chemical stimuli of their environments, such as nutrients, compounds produced by other bacteria, or compounds produced by their hosts. For example, many bacteria use signal transduction mechanisms known as quorum sensing (QS) to monitor population density and regulate resource-intensive collective processes that would not benefit individual bacteria. These pathways typically involve the production of an autoinducer ligand, whose concentration is sensed by a transcriptional regulator[1]. More recently, increasing evidence suggests that bacteria also sense and respond to physical stimuli such as those associated with contacting a rigid surface. For example, in different species specific examples of biofilm formation[2], virulence induction[3,4], swarming motility[5], and twitching motility[6] have each been shown to require bacteria to sense and respond to surface-association. QS and surface sensing have been implicated in regulating many of the same behaviors as described above, but to date have been considered largely independent processes[3,7,8].

In *Pseudomonas aeruginosa*, QS is mediated by a dense, interlinked network of to-date four known pathways: the IQS system, the quinolone system controlled by PqsR, and two LuxR/LuxI-type systems, LasR/LasI and RhlR/RhlI, respectively[9]. These systems are highly interconnected and LasR activation can induce a cascade that turns on the other systems as well[9]. When in complex with its autoinducer ligand, *N*-(3-oxododecanoyl)-homoserine lactone (3OC12-HSL), which is synthesized by LasI[10], LasR functions as a transcriptional activator. One of the factors activated by 3OC12-HSL-bound LasR is the autoinducer synthase LasI, leading to the auto-induction canonical of QS circuits[9,11–14]. In addition, LasR is required for several surface-associated behaviors, including virulence induction, but was previously thought to function in parallel to surface sensing pathways[3]. Despite its importance at the top of the *P. aeruginosa* QS hierarchy, the transcriptional regulation of *lasR* itself is relatively poorly understood[9,15]. LasR induction has to date been primarily characterized in liquid-grown planktonic cultures. In these conditions, *lasR* transcript levels increase with cell density, but do so independently of the four characterized QS pathways[9,16,17]. Furthermore, addition of saturating amounts of exogenous 3OC12-HSL ligand is not sufficient to fully induce the LasR regulon, suggesting the presence of additional unknown LasR regulators[14,18,19]. Similarly, an analysis of multiple LasR-controlled genes indicated that most of these targets are not induced at low cell density even upon addition of 3OC12-HSL[18]. In many other systems, small RNAs (sRNAs) have emerged as QS regulators[20], and LasR is known to regulate at least one sRNA, Lrs1[21]. However, the regulation of LasR expression by sRNAs or surface-association has not been examined.

Previous studies have implicated flagella, type IV pili (TFP), and PilY1 as candidate surface sensors. While flagella promote adhesion of *P. aeruginosa* in biofilm development in some contexts[22,23], virulence induction and biofilm formation share a common dependence on TFP. TFP form extracellular polymers that are actively extended and retracted by ATP-dependent motors and have been implicated as mechanosensors[4,24,25] These polymers are generally thought to be composed of a major pilin subunit, PilA, extended by the motor PilB, and retracted by the motors PilT and PilU. With respect to virulence, TFP promote attachment to host cells[26] and induce Vfr, a global regulator of virulence gene expression, in PAO1 strains[4]. In addition to PilA, *P. aeruginosa* also expresses the putative adhesin PilY1 and several minor pilin proteins with important structural and regulatory functions[27]. PilY1 is required for the ability of *P. aeruginosa* PA14

to kill amoebae host cells[3], and has been suggested to function with TFP to promote biofilm formation[7,8].

Here, we investigate the effect of surface-association on QS. We show that the QS master regulator LasR is upregulated upon surface-association, causing surface-associated cells to become more sensitive to the LasR ligand, 3OC12-HSL. Our data suggest a positive feedback system for increasing *lasR* expression that depends on the sRNA Lrs1, thereby allowing surface-associated cells to access QS-induced states that planktonic cells cannot. We also show that the increased QS sensitivity of surface-associated cells involves the TFP retraction motors and the minor pilins PilW, PilE, and PilX, but not the major pilin PilA, suggesting that different pilin forms may have distinct cellular functions. Together these results suggest that surface signaling may be an integrated system with feedbacks and cross-talk among multiple signaling pathways.

## Results

**QS master regulator LasR is upregulated on a surface**. Multiple surface-associated behaviors of *P. aeruginosa* require QS[3,28], including virulence induction and swarming motility[29]. QS and surface sensing pathways have been considered to function independently because *lasR* is not required for many of the transcriptional changes associated with surface-association. However, the significant overlap in QS and surface-dependent phenotypes suggests that these pathways may be interconnected. Consequently, we sought to determine how surface-association influences QS. We compared the mRNA abundance of the QS master regulator *lasR* in planktonic and surface-associated cells (Fig. 1a) by both RNA-seq (Fig. 1b) and qRT-PCR (Fig. 1c) in conditions that we previously showed are sufficient to induce virulence[3]. We found that *lasR* expression is upregulated ~2-fold after 1.5 h of attachment, as confirmed by both assays (Fig. 1b, c). In contrast, a well-characterized direct target of LasR, *lasI*, did not show the same surface induction in either assay (Fig. 1b and Supplementary Fig. 1).

To determine if the increase in *lasR* levels upon surface-association is a result of cells experiencing a higher effective concentration of 3OC12-HSL AI from being confined on a surface, we measured the surface-associated *lasR* induction in a *lasI* mutant, which cannot synthesize 3OC12-HSL. Expression of *lasR* still increased after surface-association in a *lasI* mutant (Fig. 1c), suggesting that the changes in *lasR* expression may be induced by surface-association itself. This hypothesis is further supported by the fact that most LasR-regulated QS genes are not upregulated upon surface-association (Supplementary Data 1), suggesting that surface-association does not merely reflect strong activation of planktonic QS signaling.

**Surface-associated cells are more sensitive to QS**. The presence of elevated *lasR* mRNA levels suggests that surface-associated cells might be hyper-responsive to 3OC12-HSL. To test this hypothesis, we examined the expression of three targets that are 3OC12-HSL/LasR-dependent but surface-independent, *lasI, rhlR*, and *pqsA*. We treated cells with 1 μM exogenous 3OC12-HSL during the 1.5 h of surface attachment, during which *lasR* increases twofold. Upon 3OC12-HSL addition, *lasI, rhlR*, and *pqsA* mRNA levels increased significantly more in surface-associated cells than in planktonic cells (Fig. 2a–c). The surface-associated hyper-induction of 3OC12-HSL targets was completely dependent on the presence of LasR as it was eliminated in a *lasR* deletion strain (Fig. 2a).

As an additional measure of the sensitized QS response, we compared the dose-response curves of surface-attached and planktonic cells to 3OC12-HSL. Using a fluorescent reporter

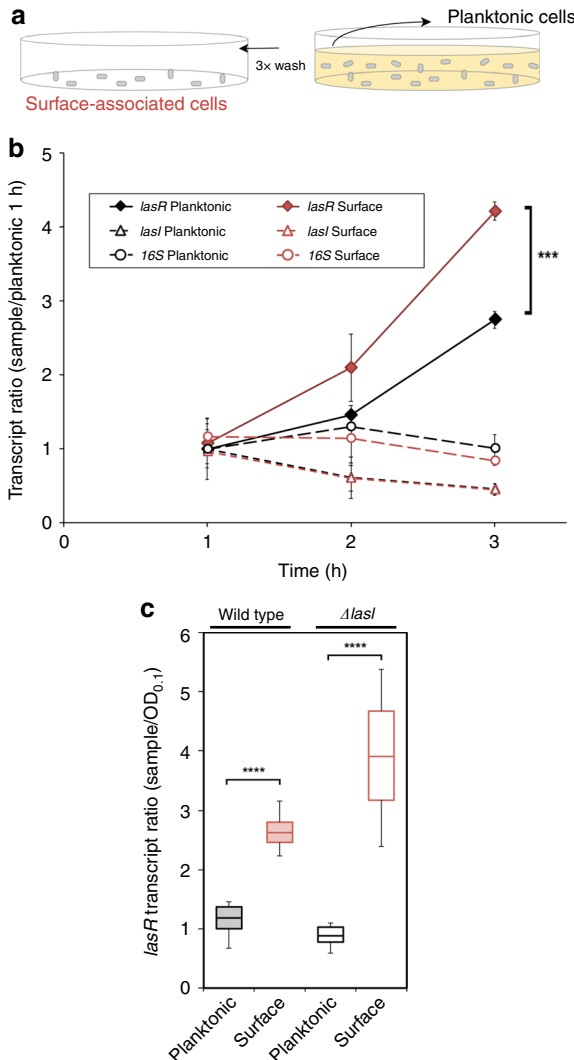

**Fig. 1** Surface association increases the expression of the QS master regulator *lasR*. **a** Diagram of the experimental procedure for obtaining surface-attached and planktonic cells. **b** Expression of *lasR* and *lasI* and control 16S from RNA-seq time course experiments normalized by the planktonic expression at 1 h ($n = 3$). The biological replicates were averaged, with the standard deviation marked with error bars. In this and all subsequent plots planktonic conditions are shown in black and surface-attached conditions are shown in red. **c** *lasR* response in a $\Delta lasI$ mutant that cannot produce 3OC12-HSL from qRT-PCR at timepoint of 1.5 h ($n = 9$). This and all subsequent qRT-PCR experiments are normalized by both 5S rRNA and OD. Error bars in plots represent standard deviation, and boxes indicate the interquartile range with the center representing the mean. A two tailed student's $t$-test was performed for comparison between samples, and significance is denoted by the asterisks (*$p < 0.01$, **$p < 0.01$, ***$p < 0.001$, ****$p < 0.0001$). Source data are provided as a Source Data file

fusion to the *lasI* promoter in a strain that cannot produce its own 3OC12-HSL, we found that the response to 3OC12-HSL saturates at significantly higher levels when cells are surface-attached (Supplementary Fig. 1A, B). These fluorescent reporter assays were performed using a glass surface, while the RNA-seq and qRT-PCR assays were performed using polysterene surfaces, indicating that the sensitization of QS is not dependent on the composition of the surface. These results also indicate that at lower 3OC12-HSL concentrations that are more likely to be physiologically relevant, there is significantly higher induction of LasR targets when cells are surface-attached (Supplementary

Fig. 1A). Analysis of the individual cell responses indicate that the QS responses are roughly normally distributed, suggesting that surface attachment increases the overall sensitivity of the population rather than causing a small subpopulation to become extremely sensitive (Supplementary Fig. 1C). Our findings that planktonic cells do not strongly induce LasR targets by 1.5 h of 3OC12-HSL treatment and have lower target induction at saturation are consistent with a previous report that high levels of 3OC12-HSL are not sufficient to fully induce most LasR targets at low cell density[18,30]. Since mRNA levels do not always reflect protein levels, we generated a LasR-FLAG fusion to directly assess LasR protein levels by Western blot. This assay confirmed that LasR-FLAG protein increases more in surface-attached cells than in planktonic cells in the presence of 3OC12-HSL (Supplementary Fig. 2).

While the results above focus on LasR targets that are more strongly activated upon surface attachment, there are also targets that appear to only become activated by LasR in the surface-associated context. For example, previous studies on QS in planktonic cells found that 3OC12-HSL does not transcriptionally induce its LasR receptor[15]. Consistent with these reports, we found that 3OC12-HSL addition had little effect on *lasR* expression in planktonic cells (Fig. 2D). In contrast, 3OC12-HSL induced *lasR* 8.5-fold more in surface-associated cells than in untreated planktonic cells (Fig. 2D). These *lasR* levels were 2-fold greater than those of stationary-phase planktonic cells at higher density ($OD_{600}$ of 4.0 for the stationary phase cells as opposed to $OD_{600}$ of 0.6 for the surface-associated cells) (Fig. 2e). These results reveal a condition-specific positive feedback loop in which 3OC12-HSL-bound LasR induces its own expression in surface-associated cells but not in planktonic cells.

The above results suggest that *lasR* induction may be an early response to surface-association. We note that while the extent of induction of LasR within 1.5 h of being introduced to a surface is modest, this induction represents a population-averaged lower limit that likely under-reports the extent of induction due to individual cells only associating with the surface for only a fraction of that timespan. For brevity, we henceforth refer to the increased QS sensitivity upon surface-association as surface-primed QS.

**Surface-primed QS requires the sRNA Lrs1.** Our findings indicate that surface-primed QS depends on *lasR*, but what factors promote *lasR* induction upon surface-association? sRNAs represent intriguing candidate regulators as they have been implicated in surface-associated behaviors, like biofilm formation and virulence, as well as in QS signaling pathways[31–34]. We thus examined surface-primed QS upon deletion of the RNA chaperone Hfq, which is required for the stability and function of many sRNAs[35]. Addition of 3OC12-HSL to a $\Delta hfq$ strain no longer showed induction of *lasR* mRNA upon surface-association (Fig. 3b).

Since the dependence of surface-primed QS on Hfq suggested the possible involvement of an sRNA, we performed RNA-seq in the conditions where cells become sensitized to QS and mapped our data set to a previously-compiled list of predicted sRNAs in *P. aeruginosa* PA14[21]. We focused on one sRNA, Lrs1, that was significantly upregulated upon surface-association (Fig. 3c), showed similar qualitative expression trends to those of *lasR* (Fig. 1b), and was previously shown to be regulated by LasR[21]. Deletion of *lrs1* reduced *lasR* transcripts to undetectable levels even upon 3OC12-HSL addition to surface-associated cells (Fig. 3d). Furthermore, constitutive overexpression of Lrs1 caused *lasR* to become expressed in planktonic cells in a manner that did not significantly increase in surface-associated cells (Fig. 3d).

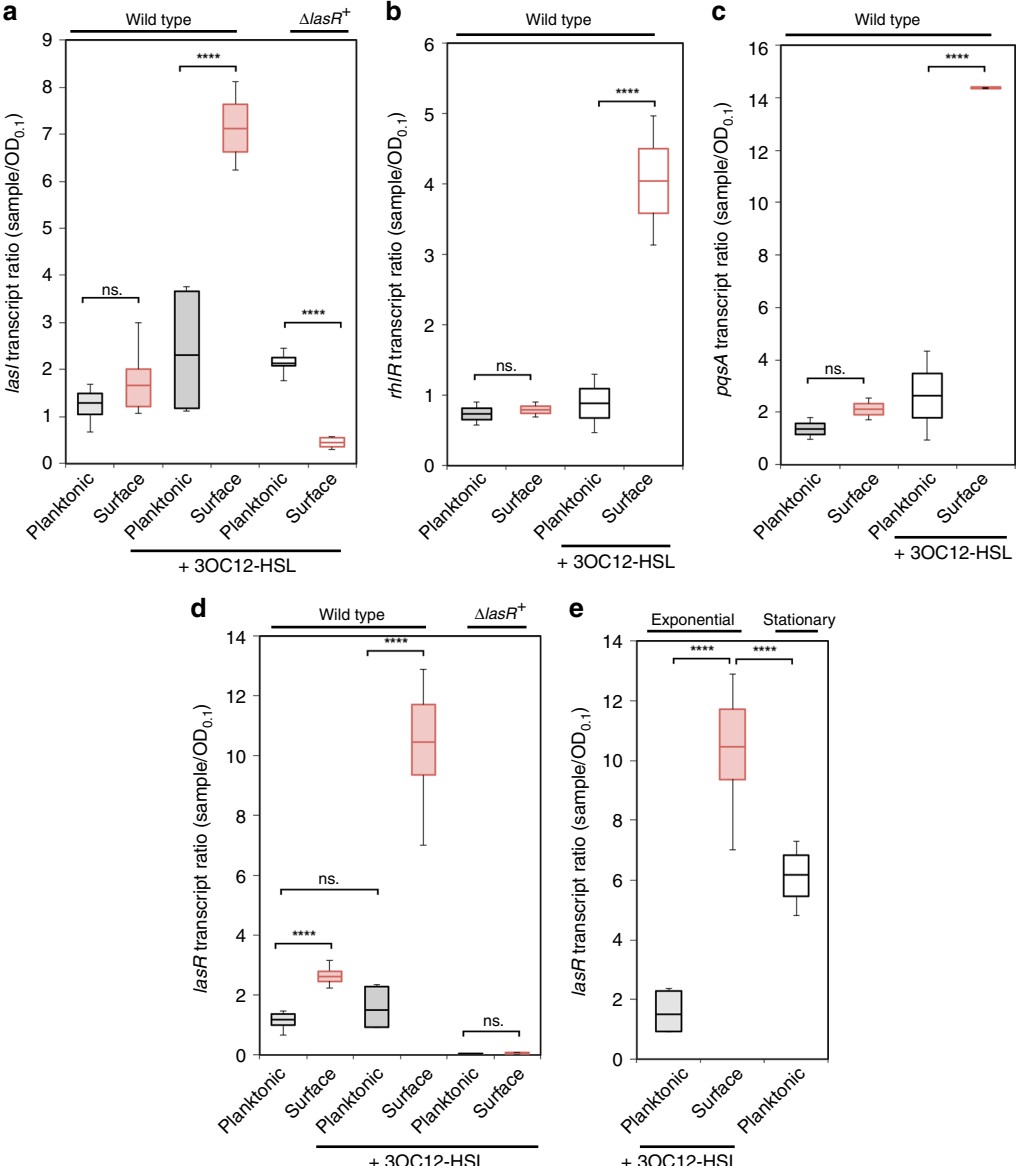

**Fig. 2** Surface-association increases QS response. qRT-PCR levels normalized to OD and 5 S rRNA for *lasI* **a**, *rhlR* **b**, *pqsA* **c**, and *lasR* **d** expression after 1.5 h treatment with 3OC12-HSL or vehicle control. [+]($ΔlasR$ mutant data collected after 3 h treatment due to decreased surface-attachment phenotype).
**e** Expression of *lasR* in exponential phase compared to stationary phase by qRT-PCR. Error bars in plots represent standard deviation, and boxes indicate 25/75 data with the center representing the mean. A two tailed student's *t*-test was performed to determine significance between samples and is denoted with asterisks ($n = 9$) (ns. = no significance, *$p < 0.01$, ****$p < 0.0001$). Source data are provided as a Source Data file

These results suggest that constitutive Lrs1 expression induces *lasR* while disrupting its normal surface-sensitive regulation.

On the chromosome, *lrs1* is encoded upstream of the promoter region of an operon encoding genes of another QS pathway, the *pqsA-E* genes that control PQS production (Fig. 3a). To differentiate whether the effects of deleting *lrs1* are due to the loss of Lrs1 itself or an indirect loss of *pqsA* promoter activity, we examined the QS response in a Δ*pqsA-E* mutant. We found that unlike Δ*lrs1*, Δ*pqsA-E* maintained *lasR* induction by 3OC12-HSL in surface-associated cells (Fig. 3e). We note that while Δ*pqsA-E* maintains surface-sensitive *lasR* induction, the absolute levels of *lasR* change in this background, consistent with previous suggestions of significant feedbacks and interconnectivity between the *P. aeruginosa* QS systems[9]. Combined with the fact that Lrs1 overexpression can induce *lasR* expression even when not expressed from its native locus, these data suggest that Lrs1 acts as a PQS-independent trans-acting regulator of *lasR*.

The requirement of Lrs1 for *lasR* induction upon surface-association suggests that Lrs1 should also be required for the QS responses downstream of LasR. We thus examined the response of the LasR target, *lasI*, to 3OC12-HSL in planktonic and surface-associated cells. These results demonstrated that Δ*lrs1* abolishes hyper-induction of *lasI* upon surface-association while Δ*pqsA-E* does not. In addition, constitutive Lrs1 overexpression induces *lasI* but renders it surface-independent (Supplementary Fig. 3). Finally, we assessed the effect of Δ*lrs1* on another QS-dependent phenotype using an Elastin-Congo Red elastase activity. As predicted by the effects of Lrs1 on LasR, Δ*lrs1* reduced elastase activities to levels comparable to those of Δ*lasR*, while constitutively overexpressing Lrs1 maintained elastase activity at or above WT levels (Fig. 3f).

A previous study demonstrated that LasR directly regulates the expression of *lrs1*[21]. Here we show that Lrs1 is also required for the expression of *lasR* (Fig. 3d). Thus, *lrs1* and *lasR* appear to be

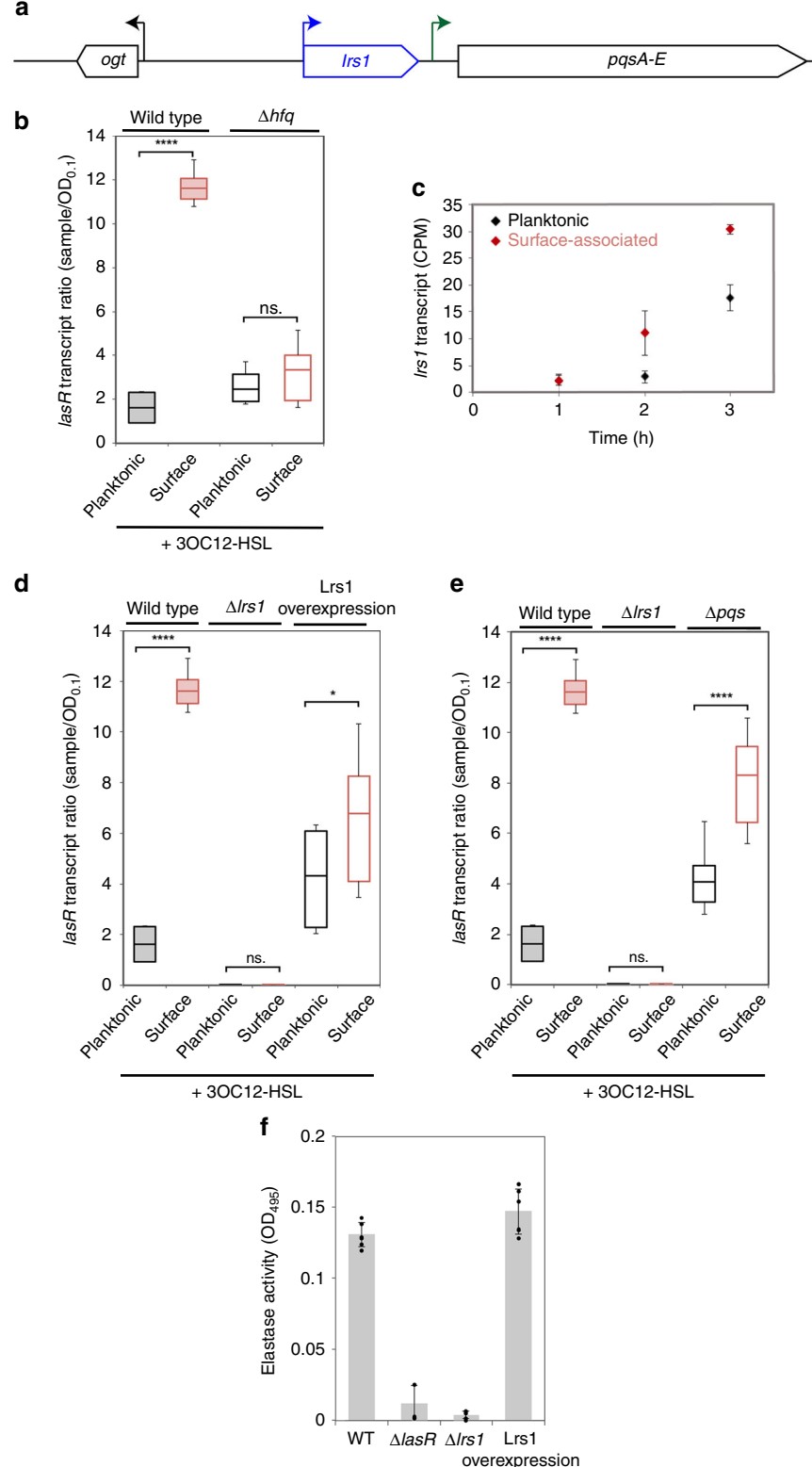

**Fig. 3** sRNA Lrs1 is required for surface-primed QS. **a** Depiction of *lrs1* location in the *P. aeruginosa* genome. **b** *lasR* transcript levels from qRT-PCR. **c** Lrs1 levels from RNA-seq data for planktonic and surface-attached cells. **d** *lasR* abundance by qRT-PCR in *Δlrs1* and upon Lrs1 overexpression. **e** *lasR* abundance by qRT-PCR in *ΔpqsA-E*. **f** Congo Red-Elastase assay comparing WT, *ΔlasR*, *Δlrs1* and Lrs1 overexpression. Individual data represented by black points (*n* = 6). Error bars in plots represent standard deviation and boxes indicate 25/75 data with the center representing the mean. A two-tailed student's *t*-test was performed to determine significance between samples and is denoted by asterisks (*n* = 9) (ns. = no significance,*p < 0.01, ****p < 0.0001). Source data are provided as a Source Data file

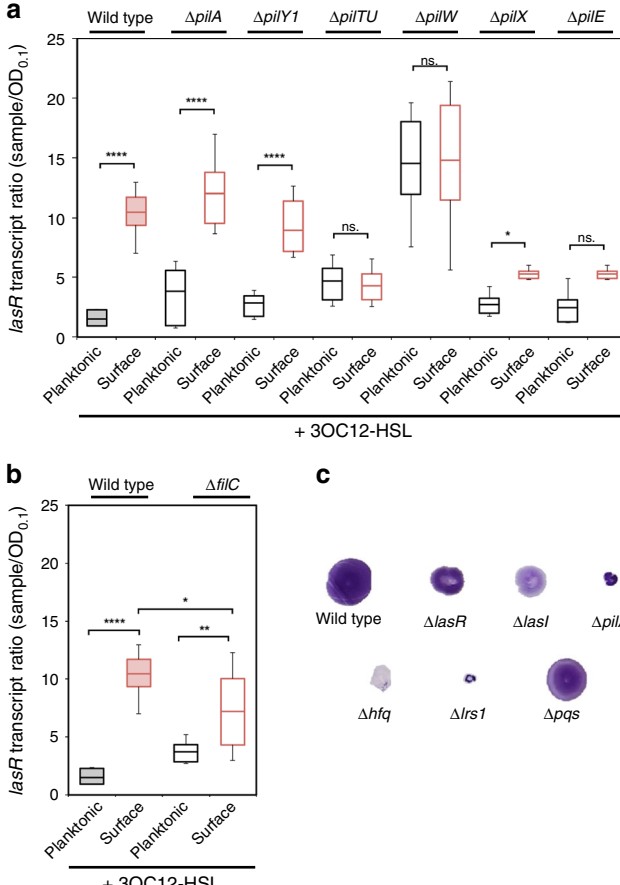

**Fig. 4** Surface-primed QS is mediated through TFP retraction motors and minor pilins. **a** *lasR* abundance as measured by qRT-PCR in TFP mutants: Δ*pilA*, Δ*pilY1*, Δ*pilTU*, Δ*pilW*, Δ*pilX*, and Δ*pilE*. **b** *lasR* abundance as measured by qRT-PCR in flagella mutant Δ*fliC*. **c** TFP-dependent twitching phenotype in QS (Δ*lasR*, Δ*lasI*) and sRNA mutants (Δ*hfq*, Δ*lrs1*) with controls Δ*pqsA-E* and Δ*pilA*. Error bars in plots represent standard deviation and boxes indicate 25/75 data with the center representing the mean. A two-tailed student's *t*-test was performed to determine significance between samples and is denoted by asterisks (*n* = 9) (ns. = no significance, *p < 0.01, **p < 0.01, ****p < 0.0001). Source data are provided as a Source Data file

interdependent, with each gene affecting the expression of the other. These data suggest a model in which upon surface-association, the induction of Lrs1 induces LasR, which positively feedbacks to yet further induce *lrs1*, generating a robustly surface-primed QS response.

**TFP retraction motors and minor pilins affect surface-primed QS.** Several systems have recently been implicated as surface sensors in *P. aeruginosa* or other bacteria. These include surface structures such as flagella, TFP and PilY1[8,23,36]. We thus tested mutants lacking these structures and their accessory factors for their involvement in surface-primed QS (Fig. 4). Surprisingly, we found that surface-primed QS was not strongly dependent on flagella (Fig. 4b), PilY1, or the major pilin PilA. However, surface-primed QS was influenced by the TFP retraction motors, PilT and PilU, and the minor pilins, PilE, PilX, and PilW (Fig. 4a). *pilTU* mutants appeared surface-blind in that they displayed low *lasR* levels regardless of whether they were planktonic or on a surface. Similarly, the *pilE* and *pilX* minor pilin mutants significantly reduced *lasR* induction upon surface attachment. In contrast,

*pilW* mutants appeared to be surface-locked in that they had high *lasR* levels in both planktonic and surface-associated conditions. Since *pilA* had no effect on *lasR* regulation, these results suggest that the minor pilins and PilTU motors have a PilA-independent role in surface signaling and that PilEX and PilW have opposing functions in this process.

Given the unexpected nature of the PilA-independent *pilW* dependence we confirmed that all of the mutants tested maintained their predicted twitching phenotypes (Supplementary Fig. 4). We also confirmed that the twitching defects of *lasR* and *lasI* can be complemented such that their defects are not due to compensatory mutations (Supplementary Fig. 5). To quantify twitching we used a plate twitching assay in which the diameter of the twitch zone serves as a measure of twitching as indicated by the small diameter of a twitching-defective *pilA* mutant control. We also examined twitching in Δ*lrs1* mutants and found that Δ*lrs1* disrupts twitching motility (Fig. 4c). This result indicates that the Lrs1-dependent surface-associated signaling pathway affects additional surface-associated behaviors beyond QS.

## Discussion

LasR can function at the top of the QS cascade in *P. aeruginosa*, but relatively little is known about the regulation of LasR itself[9]. Here we show that *lasR* is upregulated upon bacterial surface-association and that this upregulation is functional as it causes cells to respond more strongly to the LasR ligand, 3OC12-HSL. These results suggest that QS can operate in two different regimes in *P. aeruginosa*, with the same level of autoinducer synthesis causing low levels of QS activation in planktonic cells and high levels of QS activation in surface-associated cells. The fact that the system saturates at a low level in low-density planktonic cultures (higher autoinducer levels do not increase activation of planktonic cells) indicates that the QS activation achievable by planktonic cells is capped. These findings support a threshold model in which strong LasR targets that can be activated with lower LasR concentrations are induced in both planktonic and surface-associated cells, while weaker LasR targets that require higher LasR concentrations for induction are only induced in surface-associated cells.

In the future it will be interesting to determine if and how the coupled priming of surface sensing and QS benefits *P. aeruginosa* and whether similar coupling occurs in other systems. For example, the threshold for the number of community members needed to launch a successful behavior such as virulence induction could be lower if the bacteria are already on the surface of a host cell. Alternatively, the bacteria could sense one another as surfaces to hyper-activate community behaviors within aggregates like biofilms or anticipate the future accumulation of progeny upon colonizing an abiotic surface.

In this work, we observe that the induction of *lasR* upon surface association forms a positive feedback loop involving the sRNA Lrs1. It is interesting to note that sRNAs oftentimes hold crucial positions in regulatory circuits including QS, virulence and biofilm formation pathways in bacteria ranging from *Vibrio cholerae* to *Streptococcus pneumoniae*[34,37,38]. The versatility of sRNAs to act as both positive and negative regulators of gene expression, and their ability to simultaneously control multiple targets make sRNAs ideal central nodes within regulatory networks[39,40]. In addition, bacterial sRNAs differ from regulatory proteins such as transcription factors with regard to their regulation dynamics. These distinct characteristics may be relevant for the timely and precise adjustment of the gene expression profile in response to surface sensing.

Typically, bacterial sRNAs engage short, often imperfect base-pairing interactions with target mRNAs to alter the stability and/

or translation of these transcripts[41]. Whether *lasR* mRNA is directly controlled by Lrs1 through an RNA-RNA interaction in response to surface attachment remains to be determined. It is also possible that Lrs1 affects *lasR* expression indirectly by modulating another factor that in turn regulates LasR. Expression of *lasR* increases in a growth-dependent manner, but basal transcription in the absence of auto-inducers reveals that *lasR* expression is not solely controlled by QS[17]. Several factors have been identified to adjust *lasR* transcription, including positive regulation via Vfr[15] and the GacA/GacS system[42], as well as negative regulation through AlgR2[43]. It will be interesting to determine the target repertoire of Lrs1 in a future study, and to elucidate its role in *lasR* regulation.

Our findings suggest that LasR signaling is similar to that of many other QS systems in surface-associated cells. Meanwhile, the features of LasR signaling that were previously thought to be idiosyncratic (such as the lack of feedback onto *lasR* expression) could reflect the lack of the pathway's complete activation in planktonic cells. The surface-specific increase of *lasR* expression may also explain why so many surface phenotypes in *P. aeruginosa* are QS dependent.

The fundamental question of how bacteria sense the presence of a surface is also poorly understood. *P. aeruginosa* have emerged as a model system for surface sensing, with multiple distinct surface-sensing pathways. One pathway involves TFP and the Chp chemosensory system to induce cAMP and the Vfr transcription factor, which activates the expression of virulence factors[7,8]. Another pathway upregulates c-di-GMP through PilY1 and possible involvement of TFP, which influences biofilm formation[7,8]. In all previous cases where TFP have been implicated in *P. aeruginosa* surface sensing, the behavior requires the major pilin subunit, PilA[7,8]. However, our observations suggest that surface sensing mechanisms may be even more complex as we identify a surface sensing pathway that is independent of PilA yet influenced by the minor pilins PilW, PilE, and PilX, and the retraction motors PilTU. Minor pilins are generally thought to be integrated into the TFP fiber, and hypothesized to serve as adhesins to specific surfaces[44]. However, PilW may be distinct from other minor pilins as it does not bind to FimU[45] and analysis of protein levels suggests that while other minor pilins cannot be secreted in the absence of PilA, a small amount of PilW can become extracellular even without the major pilin [45].

In addition to being independent of PilA, the *pilW* response is surprising in that it appears "surface-locked", with surface-primed QS even in the planktonic state. This result suggests that PilW somehow inhibits *lasR* induction while the other minor pilins PilE and PilX stimulate *lasR* induction. There are at least two models for how *P. aeruginosa* might detect the surface in a manner that is independent of PilA, inhibited by PilW, and dependent on PilTU and PilEX. Pilin proteins often have feedback mechanisms to increase expression when the pili are active[46]. It is thus possible that intracellular PilW inhibits *lasR* induction through Lrs1 in a manner that is antagonized by the function of PilTU and PilEX. Alternatively, *P. aeruginosa* may respond to mechanical tension in surface structures, for example by PilTU causing retraction of fibers based on PilEX that are inhibited by PilW. However, for this model to be viable the minor pilins would have to form extracellular structures independently of PilA, which has not been previously reported. In any event, our data suggest that the mechanical and signaling roles of the minor pilin subunits are worthy of further investigation in future studies as they may play important roles independently of the major pilin subunit.

Regardless of the mechanistic details, our findings that surface-associated cells are more sensitive to QS compared to their planktonic counterparts have implications for how *P. aeruginosa*

coordinates the multiple behaviors in exhibits on surfaces such as biofilm formation, virulence induction, and surface motility. Previous studies demonstrated that QS is required for other surface behaviors, but transcriptional profiling suggested that this relationship can be independent[3]. Our current work provides a potential explanation for this apparent paradox. We suggest that by using an independent surface sensing pathway to sensitize themselves to QS, *P. aeruginosa* cells can effectively differentiate, allowing them to use the same QS cues already available to access states they could not access in the absence of a surface. In this sense our current work supports the previous suggestion of surface sensing and cell density forming an "AND gate"-like regulatory mechanism where *P. aeruginosa* only commits to energetically costly behaviors such as virulence induction if it is present in conditions where it is both on a surface sufficiently rigid to trigger surface sensing, and at high enough densities to trigger QS[3]. We thus effectively expand the canonical QS pathway for surface-associated cells; cells detect a surface and then prime themselves for a QS response that is only fully activated once the cells reach high cell density. While surface sensing and QS responses are coupled, they are still independent because surface-associated responses cannot be recreated by simply increasing the density of cells and QS cannot be fully activated in the absence of surface-association. Further elucidating these signaling cascades may thus respresent new ways of blocking virulence induction and biofilm formation for the development of novel anti-infective therapies.

## Method

**Bacterial strains and growth conditions**. *P. aeruginosa* strain PA14 is referred to as wild type (WT) throughout this study. All strains used in this study are listed in Table 1, all oligonucleotides are summarized in Table 2. Bacteria were routinely grown in Luria broth (LB) or LB solidified with 1.5% agar. For qRT-PCR and RNA-seq experiments, overnight cultures were diluted 1:1000 in LB and grown at 37 °C to OD$_{600}$ 0.1. Cultures were transferred to petri dishes, and incubated on an orbital shaker at 100 rpm and 37 °C. For the planktonic conditions, aliquots of culture

---

### Table 1 Bacterial strains

| Strain Name | Description | Reference |
|---|---|---|
| Wild type (WT) | *P. aeruginosa* strain PA14, clinical isolate from burn wound | 53 |
| Δ*lasR* | PA14 Δ*lasR* | 28 |
| Δ*lasI* | PA14 Δ*lasI* | 54 |
| Δ*hfq* | PA14 Δ*hfq* | 21 |
| Δ*lrs1* | PA14 Δ*lrs1* | 21 |
| Lrs1 overexpression | PA14 glmS::[P$_{A1/04/03}$-lrs1] | This work |
| Δ*pqsA-E* | PA14 Δ*pqsABCDE* | This work |
| Δ*fliC* | PA14 P$_{fro}$::yfp attB::[P$_{A1/04/03}$-mCherry] Δ*fliC* | 47 |
| Δ*pilA* | PA14 P$_{fro}$::yfp attB::[P$_{A1/04/03}$-mCherry] Δ*pilA* | 47 |
| Δ*pilC* | PA14 Δ*pilC* | 22 |
| Δ*pilD* | PA14 Δ*pilD* | 55 |
| Δ*pilE* | PA14 Δ*pilE* | 55 |
| Δ*pilTU* | PA14 Δ*pilT*/Δ*pilU* | 56 |
| Δ*pilW* | PA14 Δ*pilW* | 55 |
| Δ*pilX* | PA14 Δ*pilX* | 55 |
| Δ*pilY1* | PA14 Δ*pilY1* | 57 |
| FLAG-lasR | PA14 FLAG-lasR | This work |
| Δ*lasR* lasRI::attB | PA14 Δ*lasR* lasRI::attB | This work |
| Δ*lasI* lasRI::attB | PA14 Δ*lasI* lasRI::attB | This work |
| P$_{lasI}$-mCherry | PA14 P$_{lasI}$-mCherry::attB | This work |
| Δ*lasI* P$_{lasI}$-mCherry | PA14 Δ*lasI* P$_{lasI}$-mCherry::attB | This work |
| FLAG-lasR P$_{lasI}$-mCherry::attB | PA14 FLAG-lasR P$_{lasI}$-mCherry::attB | This work |
| S17 | *E. coli* | |

**Table 2 Oligonucleotides**

| Primer | Purpose | Sequence 5′ to 3′ |
|---|---|---|
| lrs1-1 | Overexpression construct | TTAAAGAGGAGAAATTAAGCGCCATCTCATGGGTTCGG |
| lrs1-2 | Overexpression construct | AGGAATTCCTCGAGAAGCTTCGCTGGGCGAAGCGCGAT |
| lrs1-3 | Overexpression construct | AAGCTTCTCGAGGAATTCCTG |
| lrs1-4 | Overexpression construct | CTAGAATTAAAGAGGAGAAATTAAGC |
| pqsAE-1 | Deletion construct | GATACAAAGCTTGCAAA GGCAGGCGAGACG |
| pqsAE-2 | Deletion construct | TCAGTCCAGAGGCAGCGCCTCGGTCAGGTTGGC CAATG |
| pqsAE-3 | Deletion construct | CATTGGCCAACC GACCGAGGCGCTGCCTCTGGACTGA |
| pqsAE-4 | Deletion construct | GATACAAAGCTTGCCTGCGACTCTTCCTGG |
| pqsAE-5 | Deletion construct | CCCAGTGTACTACGCAATGG |
| pqsAE-6 | Deletion construct | CCATGCAGGCGTCGCCAC |
| lasR_F | qRT-PCR | CGTACTGCCGATTTTCTGGG |
| lasR_R | qRT-PCR | AGTGCGTAGTCCTTGAGCAT |
| lasI_F | qRT-PCR | CTTTTCCGACTGTACGCTGG |
| lasI_F | qRT-PCR | AAAGCGCGATCTGGGTCTTG |
| rhlR_F | qRT-PCR | CCTGGAAAAGGAAGTGCGGC |
| rhlR_R | qRT-PCR | CCAGCCAGGCCTTGGGATAG |
| pqsA_F | qRT-PCR | GATATCGCCGACTGCCA |
| pqsA_R | qRT-PCR | GGTGGAACCCGAGGTGTATT |
| 5s_F | qRT-PCR | GAACCACCTGATCCCTTCCC |
| 5s_R | qRT-PCR | TAGGAGCTTGACGATGACCT |
| lasI-cherry-1 | Reporter construct | GAT ACA AAG CTT CCG GGT TCA CCG AAA TCT ATC TC |
| lasI-cherry-2 | Reporter construct | CCT CGC CCT TGC TCA CCA TCT TCA CTT CCT CCA AAT AGG AAG C |
| lasI-cherry-3 | Reporter construct | GCT TCC TAT TTG GAG GAA GTG AAG ATG GTG AGC AAG GGC GAG G |
| lasI-cherry-4 | Reporter construct | GAT ACA AAG CTT CTA CTT GTA CAG CTC GTC CAT GCC |
| lasRI-5 | Complementation construct | GAT ACA AAG CTT CCA TGG GAC GCC TGT TCG |
| lasRI-3 | Complementation construct | GAT ACA AAG CTT TCT GTG TCG CCG AAC TGG |

media were centrifuged, and the cell pellets were resuspended in lysis solution (TE buffer pH 8.0, 0.5 mg/mL lysozyme (Sigma), 1% SDS). For the surface attached condition, culture media was removed from the petri dish, and surface-attached cells were washed three times with DB (5 mM $KH_2PO_4$, 5 mM $Na_2HPO_4$, 2 mM $MgCl_2$, 1 mM $CaCl_2$, at pH 6.5). Lysis solution was added directly to the petri dish, and the sample was collected from the dish using a sterile cell scraper. For the RNA-Seq experiment, samples were collected at time points 1, 2, and 3 h. For qRT-PCR experiments, samples were collected at 1.5 h. 3OC12-HSL (Sigma) dissolved in DMSO was added to a final concentration of 1 μM, unless otherwise stated.

**Strain construction.** To generate an Lrs1 overexpression strain, the *lrs1* gene was amplified by PCR (primer pair lrs1-1, lrs1–2) and inserted into the pUC18-mini-Tn7 derivative, pAS08[47] (amplified by PCR using primer pair lrs1–3, lrs1–4), using Gibson Assembly[48]. The resulting construct was integrated into the T7 phage attachment site of wild-type PA14 following the procedures of Choi et al.[49]. An unmarked in-frame deletion of the *pqsABCDE* operon was generated using two-step allelic exchange as described before[50]. The regions 500-bp upstream and downstream of the operon were amplified by PCR (pqsAE-1, pqsAE-2) and (pqsAE-3, pqsAE-4), respectively, fused through overlap extension PCR (SOE-PCR), and ligated into the HindIII site of plasmid pEXG2. Deletions were confirmed using by colony PCR (using primer pair pqsAE-5, pqsAE-6). To generate the FLAG-tagged *lasR* strain, a roughly 800-bp sequence containing 3xFLAG motif fused to the N-terminus *lasR* gene was synthesized commercially. This fragment was cloned into the HindIII site of pEXG2, and *FLAG-lasR* was integrated onto the chromosome through allelic exchange. Proper insertion was confirmed by sequencing. To generate the *lasRI* complementation strain, the *lasRI* genes were amplified from gDNA using primer pair (lasRI-5, lasRI-3) and cloned into the HindIII site of the mini-CTX plasmid. The CTX-2 construct was integrated into the chromosome of *lasR* and *lasI* mutants at the *attB* phage attachment site. To generate the P*lasI*-mCherry promoter fusion, a ~300 bp fragment upstream of the *lasI* gene was amplified from gDNA using primer pair (lasI-cherry-1, lasI-cherry-2). This fragment was fused by SOE-PCR to a promoterless *mCherry* fragment, amplified from plasmid pAS08 using primer pair (lasI-cherry-3, lasI-cherry-4). The resulting fusion was cloned into mini-CTX-2 and integrated onto the chromosome at the *attB* site.

**RNA isolation.** Total RNA was from isolated from surface-attached and planktonic cells as previously described[3]. In brief, planktonic cells were collected by pelleting 600 μL of the liquid culture in the dish. The supernatant was removed and immediately lysed with lysis solution (TE buffer pH 8.0, 0.5 mg/mL lysozyme (Sigma), 1% SDS). Surface-associated cells were isolated by washing the dish three times with DB (5 mM $KH_2PO_4$, 5 mM $Na_2HPO_4$, 2 mM $MgCl_2$, 1 mM $CaCl_2$, at pH 6.5). Lysis solution was added to isolate surface-attached cells from the petri dish, and the sample was collected from the dish using a sterile cell scraper within a few

minutes of the experimental time point. Total RNA was isolated from all samples collected in lysis solution using hot phenol extraction. Genomic DNA was removed from all total RNA samples by DNase I digestion.

**qRT-PCR prep and analysis.** For each experimental condition, cDNA was synthesized from 1.5 μg total RNA using Superscript III (Invitrogen) following the manufacturer's recommendations. cDNA was mixed with gene-specific primer sets (Tables 2) and 2× PerfeCTa SYBR Green FastMix Low Rox (Quanta Bio), and run on a Viia7 384-Well Fast Real Time PCR System (Thermo Fisher) in the Princeton Genomics Core Facility. For each of the three biological replicates collected per sample, four technical replicates were run. CT values were converted to fold-change through the Pfaffl Method[51]. 5 S rRNA was used as an internal control, and the control condition present on each plate was RNA prepared from planktonic wild-type cells at $OD_{600}$ of 0.1. Each biological replicate was then averaged, and the average and standard deviation of the three biological replicates was plotted.

**RNASeq library prep and analysis.** Libraries for Illumina Sequencing of cDNA were constructed by the Princeton Genomics Core Facility. The integrity of total RNA samples (three biological replicates) was assessed on a Bioanalyzer 2100 using RNA 6000 Pico chip (Agilent Technologies, CA). For each sample, 0.5 μg of RNA were treated with the Ribo-Zero Bacteria rRNA Removal Kit (Illumina, CA) to deplete the majority of ribosomal RNAs. The RNA was fragmented and converted to cDNA libraries using the PrepX RNA-seq library kit (Takara Bio USA) on the automated Apollo 324™ NGS Library Prep System (Clontech, CA). Library-specific barcodes were introduced to allow for multiplex sequencing. The RNA-seq libraries were examined on Agilent Bioanalyzer DNA High Sensitivity chips for size distribution, quantified using a Qubit fluorometer (Invitrogen, CA), and pooled at equal molar amounts. The library pools were denatured and sequenced on Illumina HiSeq 2500 Rapid flowcells as single-end 75 nt reads according to the manufacturer's protocol. Raw sequencing reads were filtered by the Illumina HiSeq Control Software, only the Pass-Filter (PF) reads were de-multiplexed allowing one mismatch, and used for further analysis.

Sequence files from the facility were processed with customized Python scripts from our lab (which are available on request), and were aligned to the *P. aeruginosa* UCBPP-PA14 genome using Bowtie 2[52]. Raw counts were divided by total number of reads per sample and converted to counts per million (CPM) for comparison. CPM for each biological replicate were averaged, and the standard deviation is indicated by error bars. Data in plots were normalized by the planktonic counts at 1 h for each gene for comparison.

**Western blot analysis.** For western blot analysis, cultures were grown in PS:DB media, which consists of DB and 10% (v/v) PS medium (10 g L⁻¹ Special Peptone (Oxoid), 7 g L⁻¹ Yeast Extract (Oxoid), 10 mM $KH_2PO_4$, 0.45 M $Na_2HPO_4$, 15 g L⁻¹ glucose, 20 nM vitamin B12, 180 nM Folic Acid, pH 6.5). Overnight cultures of

*FLAG-pqsA* were diluted 1:100 in PS:DB and grown to $OD_{600nm} = 0.5–0.6$ at 37 °C with shaking (250 rpm). Cultures were transferred to 22 × 22 cm culture dishes and incubated for 1 h at 37 °C on an orbital shaker (80 rpm). Planktonic samples were isolated by aspirating the culture media. Surface-attached cells were washed twice in PBS and removed from the surface using a cell scraper. Cells were resuspended in loading buffer to equal concentrations, based on $OD_{600nm}$, and seperated on a 10% SDS-PAGE. Blots were probed using DYKDDDDK Tag (D6W5B) Rabbit monoclonal antibody (14793 S, Cell Signaling Technologies, Danvers, MA) at a 1:1000 dilution and a polyclonal antibody targeting the $\sigma^{70}$ subunit of RNA polymerase, provided by the Bassler lab, at 1:5000 dilution. Primary antibody was detected with anti-rabbit IgG, HRP-linked secondary antibody (7074 S, Cell Signaling Technologies, Danvers, MA) at 1:3000 dilution.

**Analysis of fluorescent promoter fusions**. Overnight cultures of *ΔlasI* expressing $P_{lasI}$-*mCherry::attB* or $P_{pqsA}$-*mCherry::attB* fluorescent fusions were diluted 1:100 in PS:DB and incubated at 37 °C with shaking (250 rpm). Various amounts of 3OC12 HSL was added when cultures reached $OD_{600nm} = 0.2$. When cultures reached $OD_{600nm} = 0.5–0.6$, cultures were transferred to glass-bottom petri dishes (Mattek Corporation, Ashland, MA) and incubated for 1 h at 37 °C on an orbital shaker (80 rpm). Planktonic cells were isolated by aspirating the culture media. Surface-attached cells were washed twice in PBS. Samples were covered with 1% agar pad and imaged using fluorescent microscopy.

**Twitching assays**. A toothpick was used to transfer cells from a solid plate to a 1.5% LB agar pad. Plates were incubated in a humid box at 30 °C for 48 h. Twitching diameters were visualized by either incubating plates for 30 min in developer solution (50% methanol, 10% glacial acetic acid in $H_2O$), or by removing agar pads and incubating plates with 1% crystal violet in water for 10 min. Plates were washed with water prior to imaging.

**Congo red elastase assay**. Cells were grown at 37 °C overnight in LB and collected by centrifugation at 16,000 × *g* for 1 min. Supernatant was passed through a 0.22 μm filter, and 100 μL of supernatant was mixed with 900 μL of 10 mM $Na_2HPO_4$ and 10 mg elastin-Congo red substrate (Sigma-Aldrich). After incubation for 2 h at 37 °C, and the samples were centrifuged at 16,000 × *g* for 10 min. The optical density of the supernatant was measured at $OD_{495}$. Each biological sample had three technical replicates.

**Reporting summary**. Further information on research design is available in the Nature Research Reporting Summary linked to this article.

## Data avaliability

The RNA sequencing data have been deposited in the SRA database at NCBI with accession code PRJNA528963.

## Code availability

Customized Python scripts used to process RNA sequencing data are available on request.

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

## Acknowledgements

We wish to thank members of the Gitai and Shaevitz labs for helpful discussions and comments on the paper, the Bassler and Lory labs for strains and reagents, and Albert Siryaporn for technical assistance. Z.G. is supported by an NIH Pioneer Award DP1-AI124669 and an award from the Princeton Catalysis Initiative.

## Author contributions

Experiments were designed and analyzed by S.C., G.V., K.F., and Z.G., experiments were performed by S.C. and G.V., and the paper was written by S.C., G.V., and Z.G.

## Additional information

**Competing interests:** The authors declare no competing interests.

