## [Peer Review File · Nature Communications]

Reviewers' comments:

Reviewer #1 (Remarks to the Author):

The manuscript by Chuang et al, is investigating the effects of surface association on quorum sensing pathways, with specific emphasis on the master regulator, LasR. The authors show that LasR is upregulated upon association with the surface, and that this upregulation causes the cells to become hyper-sensitive to the LasR inducer – 3OC21HSL. They showed using a sRNA knockout that LasR induction is dependent upon Lrs1, a previously known target of LasR, suggesting this sRNA has a critical role in quorum sensing. Finally, they showed that the surface response is dependent upon a subset of pilins expressed by pseudomonas.

Overall, I think the paper is quite interesting, and the authors have data that shows a dramatic rearrangement upon surface adhesion. This manuscript will add to the growing literature emphasizing the importance of mechanical stress effecting clinically relevant bacterial models. I think the paper will be interesting to numerous researchers.

I have one main issue in regards to the interpretation of the data. I am confused about the model that Lrs1 induces LasR. A couple of points illustrate my confusion.

- (1) In figure 3C, Lrs1 increases in both surface and planktonic cells, but that is not reflected in the LasR induction. At 3 hours post adherence, the planktonic cells show a 15x increase in the Lrs1 transcript levels, but the levels of lasR go down during that time (Fig 1B).
- (2) If Lrs1 induces lasR, why does the Lrs1 overexpression on the surface with autoinducer only half the level of the WT (Fig 3D).
- (3) The PQS data is confusing, why is that changing the lasR expression at all? Is the deletion of that operon leading to Lrs1 overexpression? The authors should test via qPCR or add a reference if this is known in the field (Fig 3E). Similarly, the authors could measure the total Lrs1 in their overexpression vector.

I am willing to believe there are hidden factors in the models that drive my confusion, but the authors should make note of those points at the very least.

Smaller issues:

I'm not sure how the twitching phenotype data is relevant. The pilA knockout shows a more dramatic phenotype than the lasR or lasI knockouts, but is not involved with the lasR regulation. If the authors want to emphasize that lasR has an effect on twitching, those mutants muddy the image. As a non-expert in twitching, it was also hard for me to see what I was supposed to get from the data. Is PQS knockout the same? It looks different around the edges as compared to WT. Is it purely the size? Please elaborate.

Figure 1 is not clear. Please make Fig 1 A larger to emphasize the protocol. Fig 1 B is not labelled, and this is one cornerstone of the paper! I assume **** is $p < .0001$, but that's not in the legend. Is Figure 1B with the autoinducer? In the text it says comparing mRNA abundance by both RNAseq and qRT-PCR, but then one has the autoinducer, and one doesn't? In all the box plots, are the boxes the 25/75 data, or the 10/90? Is the center the median or the mean?

The authors mention saturation in planktonic cultures several times, but I think it would be useful to have a dose-response generated for the surface adhered cells. It is unclear if these cells are saturated. Though it's not critical to the conclusions of the paper, it would help solidify their claims that surface enhancement enables a new state of QS.

The hfq knockout is known to globally affect transcript stability. The authors used it to justify that a small RNA may be involved, but the decrease in lasR could also be due to direct hfq effects.

In the materials and methods, the authors describe the RNA collection protocol, but it would be useful to include something about the timing. For other labs trying to replicate, it should be clear that the RNA harvesting should be done as quickly as possible since bacteria can dramatically change their transcriptome within minutes.

For the knockout and overexpression cells, are all of the RNA measurements normalized to OD 0.1, or are they to an internal gapdh control? If it's to the OD, it would be useful to know if the cells are the same morphology and growth rates as compared to the WT.

Can you comment on if there's potential meaning in the lasI knockout showing even greater surface sensitivity as compared to the WT (Fig 1C)?

In table S1, the lrs1 gene is not listed. Was it not part of the LasR regulon?

Is the RNAseq data deposited on a public database?

Can you include more details on the library prep for RNAseq?

Reviewer #2 (Remarks to the Author):

The authors are interested in investigating the effect of surface-association on quorum sensing. The authors hypothesize that there is at least one other unknown regulator of LasR. The regulation of LasR by sRNAs or surface association has not been examined.

The authors show that LasR is upregulated upon surface association. The authors also show that the surface associated QS requires the sRNA Lrs1.

Overall, I think the authors take a novel approach to investigating quorum sensing. They present a nice story and have some exciting results. There are some definite areas for improvement.

Major Comments:

1. It was a bit difficult to evaluate the manuscript because I found that the figure legends were lacking specific details (see minor comments) and there were places in the manuscript that lacked a reference to a Figure. The authors should spend some time correcting these errors.
2. The authors tested the effect of delta Lrs-1 on twitching. Did the authors consider tested the strain overexpressing lrs1? This may strengthen support for the role of lrs1.
3. The model in Fig S4 depicts changes in binding affinity of LasR with autoinducer for promoters. The manuscript does not investigate binding affinities but rather investigates the role of the small RNA Lrs-1 on lasR expression, as well as possible surface sensing mechanisms involved in the QS response. It would perhaps be more beneficial to the readers to include the results of experiments centered on surface association in the model.
4. Details in the Methods are lacking. For example, how were the qRT-PCR data analyzed? What

normalizing genes were used? For RNA-Seq analyses, the authors should be more specific. For example, how was the differential expression analyses performed? These are necessary in order to be able to appropriately evaluate the statistics.

Minor Comments:

1. Figure 1 and Figure 1 legend (page 16); Labels are missing. Please define the triangles and diamonds as well as the black and red lines. Panel (D) is indicated in the legend. Should this be panel (C)?
2. Figure 2 and Figure 2 legend: This figure and legend are confusing. The legend states the wild-type cells are shaded boxes and mutant cells are empty, however in panels (B) and (C) wild-type is indicated above the figure. Since each of the panels represent evaluate of a single gene, then perhaps it might be better to say *lasI* (A), *rhIR* (B), *pqsA* (C) and *lasR* (D)?
3. Page 7; I didn't find Table 1. Are the authors referring to Table S1?
4. Page 11, line 2; Please indicate figure here.
5. Page 11; I can not seem to find a reference to Figure 4B.
6. References: Many of the bacterial names are not italicized.

Reviewer #3 (Remarks to the Author):

In *P. aeruginosa*, the addition of exogenous 3-oxo-C12-HSL to low density cultures immediately induces the expression of some but not other quorum sensing dependent genes. For some QS target genes, this is known to depend on the availability of additional regulatory elements. In the present paper, the authors provide some interesting experimental data to suggest that the *P. aeruginosa* quorum sensing 'master regulator' gene *lasR*, is: (a) 'hypersensitively' up-regulated in polystyrene petri dish surface-associated bacterial cells compared with planktonic cells, (b) this surface dependent *lasR* up-regulation is dependent on the sRNA, *Lsr1* and that (c) the type 4 pilus protein retraction motors and minor pilins affect surface-primed levels of *lasR* expression.

While the data on *lasR* expression are interesting, the 'hypersensitivity' does not seem to have a downstream consequence for the 3-oxoC12-HSL synthase gene, *lasI* i.e. autoinduction nor the quorum sensing virulence/biofilm target structural genes as the RNASeq data shown in Table 1 illustrates. Hence the physiological consequences of *lasR*-upregulation for bacterial QS behaviour on a surface are not clear. The term 'hypersensitive' is also misleading in the sense that the experiments demonstrating this use a single high (saturating) concentration of exogenous 3-oxo-C12-HSL (1 microM) when physiological *LasR* activation requires only low nanomolar concentrations of 3-oxo-C12-HSL.

Although the links between *Lsr1*, *LasR* and the minor pilins are intriguing and the manuscript contains some interesting ideas about why priming of QS on a surface might be advantageous for *P. aeruginosa*, there is little direct mechanistic data on how *Lsr1* controls *lasR* expression or indeed how the pilin proteins control *lasR* expression in the presence of saturating 3-oxoC12-HSL.

Specific Points

1. The experimental set-up for the surface-associated vs planktonic uses a polystyrene petri dish. This raises some further questions. Is polystyrene a relevant surface for *P. aeruginosa* virulence/biofilms? Does the surface induction of *lasR* only occur on polystyrene? From the methods section, it is not clear how the surface-associated and planktonic cells were separated. Were the plates washed to remove weakly attached cells? Presumably there were far more planktonic cells than surface attached cells? Can the authors provide information on the numbers of surface associated and planktonic cells used for the subsequent transcriptome experiments? Do the cells have to be firmly attached to the surface to induce *lasR* expression or only transiently associated? Since the RNASeq/qPCR experiments only provide averages over the population, the

use of a fluorescently labelled reporter gene fusion or flow cytometry would enable direct observation of the two cell populations to back up the transcriptome data and to determine whether the 'hypersensitive' expression of *lasR* was stochastic. Since transcriptional changes do not always manifest as translational changes, the paper could be strengthened by direct measurement of changes in LasR by Western blotting or ELISA type assays.

2. The authors use a number of *P. aeruginosa* mutants but none are genetically complemented to rule out secondary mutations. Previous work linking type IV twitching motility and quorum sensing was subsequently shown to be a consequence of compensatory mutations (Beatson et al 2002 *J. Bact.*). Fig. 4C is of concern as the *lasR* and *lasI* deletion mutants show reduced twitching compared with the wild type.

3. The symbols in Fig. 2B (legend) are not explained. Fig. 2C (not 2D as in legend) is missing a 'no 3-oxoC12-HSL' control.

4. The term 'binding affinity' is used in Fig. S4 but it is not clear what this is referring to?

Reviewers' comments:

Reviewer #1 (Remarks to the Author):

The manuscript by Chuang et al, is investigating the effects of surface association on quorum sensing pathways, with specific emphasis on the master regulator, LasR. The authors show that LasR is upregulated upon association with the surface, and that this upregulation causes the cells to become hyper-sensitive to the LasR inducer – 3OC21HSL. They showed using a sRNA knockout that LasR induction is dependent upon Lrs1, a previously known target of LasR, suggesting this sRNA has a critical role in quorum sensing. Finally, they showed that the surface response is dependent upon a subset of pilins expressed by pseudomonas.

Overall, I think the paper is quite interesting, and the authors have data that shows a dramatic rearrangement upon surface adhesion. This manuscript will add to the growing literature emphasizing the importance of mechanical stress effecting clinically relevant bacterial models. I think the paper will be interesting to numerous researchers.

I have one main issue in regards to the interpretation of the data. I am confused about the model that Lrs1 induces LasR. A couple of points illustrate my confusion.

(1) In figure 3C, Lrs1 increases in both surface and planktonic cells, but that is not reflected in the LasR induction. At 3 hours post adherence, the planktonic cells show a 15x increase in the lrs1 transcript levels, but the levels of lasR go down during that time (Fig 1B).

We apologize for any confusion, but the levels of *lasR* actually increase similarly to those of *lrs1* (black diamonds in Fig. 1B). We have attempted to clarify the similarity in the increases in *lrs1* and *lasR* expression in the text on lines 216-219.

(2) If Lrs1 induces lasR, why does the Lrs1 overexpression on the surface with autoinducer only half the level of the WT (Fig 3D).

We thank the reviewer for raising this interesting question. We interpret the significant increase in *lasR* induction in planktonic cells when comparing Lrs1 overexpression and WT cells to indicate that Lrs1 is inducing LasR, but in a manner that makes it no longer surface-sensitive (since that induction is not further elevated in surface cells). This would be consistent with Lrs1 inducing *lasR* constitutively, which disrupts the normal surface-induced response. We have clarified this point on lines 223-224.

(3) The PQS data is confusing, why is that changing the lasR expression at all? Is the deletion of that operon leading to lrs1 overexpression? The authors should test via qPCR or add a reference if this is known in the field (Fig 3E). Similarly, the authors could measure the total lrs1 in their overexpression vector.

PQS is itself a quorum sensing factor and the quorum sensing pathways of *P. aeruginosa* are highly interconnected with multiple feedback systems. Here the only conclusion we draw from the delta-pqs data is that the loss of PQS looks very different from the loss of Lrs1, confirming that the loss of Lrs1 cannot be a secondary consequence of the loss of pqs (a concern since Lrs1 is near the pqs promoter). The fact that *lasR* is still surface-sensitive in the absence of pqs confirms this point. The question of why the loss of pqs does not look exactly like WT is an interesting quorum sensing regulation question, but beyond the scope of our current work. We have added a sentence

explaining this point and adding a reference to known QS feedbacks on lines 231-234.

I am willing to believe there are hidden factors in the models that drive my confusion, but the authors should make note of those points at the very least.

We appreciate these helpful suggestions and have made the recommended clarifications noted in the sections above.

Smaller issues:

I'm not sure how the twitching phenotype data is relevant. The *pilA* knockout shows a more dramatic phenotype than the *lasR* or *lasI* knockouts, but is not involved with the *lasR* regulation. If the authors want to emphasize that *lasR* has an effect on twitching, those mutants muddy the image. As a non-expert in twitching, it was also hard for me to see what I was supposed to get from the data. Is PQS knockout the same? It looks different around the edges as compared to WT. Is it purely the size? Please elaborate.

Our goal with the twitching data was to show that *Irs1* does not only affect surface-sensitive QS, but also affects other surface-sensitive responses of *P. aeruginosa*. Thus, our hope was that the reader would see that the *lasR* and *Irs1* knockouts reduce twitching relative to WT. *pilA* mutants serve merely as a control for the assay, showing what happens when twitching is eliminated altogether. Specifically, as other groups have shown previously, the diameter of the twitching zone serves as a quantitative measure of twitching, as shown by the small diameter of the twitching-defective *pilA* mutant control and large diameter of the WT twitching-competent control. We clarify these points in the text in the paragraph that starts on line 276.

Figure 1 is not clear. Please make Fig 1 A larger to emphasize the protocol. Fig 1 B is not labelled, and this is one cornerstone of the paper! I assume **** is $p < .0001$, but that's not in the legend. Is Figure 1B with the autoinducer? In the text it says comparing mRNA abundance by both RNAseq and qRT-PCR, but then one has the autoinducer, and one doesn't? In all the box plots, are the boxes the 25/75 data, or the 10/90? Is the center the median or the mean?

We agree that Fig 1B is the cornerstone of the paper and appreciate the reviewer's help in increasing its clarity. We have incorporated all of the requested suggestions.

The authors mention saturation in planktonic cultures several times, but I think it would be useful to have a dose-response generated for the surface adhered cells. It is unclear if these cells are saturated. Though it's not critical to the conclusions of the paper, it would help solidify their claims that surface enhancement enables a new state of QS.

This is a great suggestion. We have performed the dose response curve and found the predicted result that the LasR target, *lasI* saturates at lower levels in planktonic cells than in surface cells. Importantly, this was done in a delta-*lasI* background so the only 3OC12-HSL available was the exogenously-added molecule. This significantly strengthens our conclusion and we have added these results to the text, beginning on line 161, and show the new data in Fig. S1.

The *hfq* knockout is known to globally affect transcript stability. The authors used it to justify that a small RNA may be involved, but the decrease in *lasR* could also be due to direct *hfq* effects.

The reviewer is formally correct that the role of Hfq could be independent of sRNAs, but that is why we merely used this result to justify searching for the relevant sRNA, Lrs1. We have added a qualifier on line 198 to indicate that the hfq result only suggests the possible role of sRNAs, which we then go on to formally test in the subsequent experiments.

In the materials and methods, the authors describe the RNA collection protocol, but it would be useful to include something about the timing. For other labs trying to replicate, it should be clear that the RNA harvesting should be done as quickly as possible since bacteria can dramatically change their transcriptome within minutes.

We have made the requested additions.

For the knockout and overexpression cells, are all of the RNA measurements normalized to OD 0.1, or are they to an internal gapdh control? If it's to the OD, it would be useful to know if the cells are the same morphology and growth rates as compared to the WT.

We adjusted the methods section to clarify that we used a gapdh control.

Can you comment on if there's potential meaning in the lasI knockout showing even greater surface sensitivity as compared to the WT (Fig 1C)?

We are reluctant to draw significant conclusions from this relatively subtle difference, but this result could reflect the presence of the significant number of feedbacks known to be present in the QS network.

In table S1, the Lrs1 gene is not listed. Was it not part of the LasR regulon?

Table S1 includes only the protein-coding genes regulated by LasR. Lrs1 and other sRNAs are thus not included here.

Is the RNAseq data deposited on a public database?

Yes, it has been uploaded to the NCBI-SRA database and an accession number added to the text. The data will be released upon publication.

Can you include more details on the library prep for RNAseq?

We have expanded the methods section to include these details.

Reviewer #2 (Remarks to the Author):

The authors are interested in investigating the effect of surface-association on quorum sensing. The authors hypothesize that there is at least one other unknown regulator of LasR. The regulation of LasR by sRNAs or surface association has not been examined.

The authors show that LasR is upregulated upon surface association. The authors also show that the surface associated QS requires the sRNA Lrs1.

Overall, I think the authors take a novel approach to investigating quorum sensing. They present a nice story and have some exciting results. There are some definite areas for improvement.

Major Comments:

1. It was a bit difficult to evaluate the manuscript because I found that the figure legends were lacking specific details (see minor comments) and there were places in the manuscript that lacked a reference to a Figure. The authors should spend some time correcting these errors.

We thank the reviewer and have added details to the figure legends and methods sections.

2. The authors tested the effect of delta Lrs-1 on twitching. Did the authors consider tested the strain overexpressing lrs1? This may strengthen support for the role of lrs1.

The Lrs1 overexpression strain looked like WT with respect to twitching. This result could either mean that the overexpression strain does not induce enough Lrs1 to affect twitching or that something else is going on. Since it neither strengthens nor weakens our argument we have not included it in the manuscript.

3. The model in Fig S4 depicts changes in binding affinity of LasR with autoinducer for promoters. The manuscript does not investigate binding affinities but rather investigates the role of the small RNA Lrs-1 on lasR expression, as well as possible surface sensing mechanisms involved in the QS response. It would perhaps be more beneficial to the readers to include the results of experiments centered on surface association in the model.

We thank the reviewer for pointing out correctly that the model in Fig. S4 emphasizes transcription factor binding affinities rather than the emphasis of the paper. We have thus removed this model as it appears to confuse rather than clarify our message.

4. Details in the Methods are lacking. For example, how were the qRT-PCR data analyzed? What normalizing genes were used? For RNA-Seq analyses, the authors should be more specific. For example, how was the differential expression analyses performed? These are necessary in order to be able to appropriately evaluate the statistics.

We agree and have expanded these sections.

Minor Comments:

1. Figure 1 and Figure 1 legend (page 16); Labels are missing. Please define the triangles and diamonds as well as the black and red lines. Panel (D) is indicated in the legend. Should this be panel (C)?

2. Figure 2 and Figure 2 legend: This figure and legend are confusing. The legend states the wild-type cells are shaded boxes and mutant cells are empty, however in panels (B) and (C) wild-type is indicated above the figure. Since each of the panels represent evaluate of a single gene, then perhaps it might be better to say lasI (A), rhlR (B), pqsA (C) and lasR (D)?

3. Page 7; I didn't find Table 1. Are the authors referring to Table S1?

4. Page 11, line 2; Please indicate figure here.

5. Page 11; I can not seem to find a reference to Figure 4B.

6. References: Many of the bacterial names are not italicized.

We thank the reviewer for these suggestions and have incorporated all of them.

Reviewer #3 (Remarks to the Author):

In *P. aeruginosa*, the addition of exogenous 3-oxo-C12-HSL to low density cultures immediately

induces the expression of some but not other quorum sensing dependent genes. For some QS target genes, this is known to depend on the availability of additional regulatory elements. In the present paper, the authors provide some interesting experimental data to suggest that the *P. aeruginosa* quorum sensing 'master regulator' gene *lasR*, is: (a) 'hypersensitively' up-regulated in polystyrene petri dish surface-associated bacterial cells compared with planktonic cells, (b) this surface dependent *lasR* up-regulation is dependent on the sRNA, *Lsr1* and that (c) the type 4 pilus protein retraction motors and minor pilins affect surface-primed levels of *lasR* expression.

While the data on *lasR* expression are interesting, the 'hypersensitivity' does not seem to have a downstream consequence for the 3-oxoC12-HSL synthase gene, *lasI* i.e. autoinduction nor the quorum sensing virulence/biofilm target structural genes as the RNASeq data shown in Table 1 illustrates.

For the RNASeq experiment (Table 1), we specifically chose to study cells at relatively low ODs, to avoid complications from cells entering stationary phase, etc. Therefore, the fact that we do not see a significant change in *lasI* induction in the absence of exogenous 3OC12-HSL in these experiments is likely due to the relatively low amounts of autoinducer accumulated at this point. This does not mean that the hypersensitivity is not physiological, but rather supports our point that it is relevant when cells accumulate significant autoinducer levels, whose targets become expressed at higher levels when cells are on a surface. As discussed in the next section we also further support this claim with a dose-response curve now shown in Fig. S2.

Hence the physiological consequences of *lasR*-upregulation for bacterial QS behaviour on a surface are not clear. The term 'hypersensitive' is also misleading in the sense that the experiments demonstrating this use a single high (saturating) concentration of exogenous 3-oxo-C12-HSL (1 microM) when physiological *LasR* activation requires only low nanomolar concentrations of 3-oxo-C12-HSL.

Here we were using the term "hypersensitive" to indicate a higher sensitivity to the same level of autoinducer, which we believe is a valid use of this term. Nevertheless, the reviewer raises a valid concern about whether this hypersensitivity is only relevant at extraordinarily high autoinducer levels. We thus performed a dose-response curve comparing the responsiveness of *lasI* promoter fusions to 3OC12-HSL in planktonic and surface-attached cells (new Fig. S1). These results show that even at lower concentrations of 3OC12-HSL, there is still a significant increase in the sensitivity of the cells to autoinducer. Finally, we note that while some "strong" *LasR* targets like *LasI* only require low nM 3OC12-HSL levels for activation, many other *LasR* targets require much higher autoinducer levels or are never activated in a planktonic state (see for example Whiteley et al., 1999). This supports our model that there is *some* *LasR* activation in planktonic cells and that this activation is strengthened upon surface association. We have made changes to clarify these points, which include the addition of the paragraph at line 161.

Although the links between *Lsr1*, *LasR* and the minor pilins are intriguing and the manuscript contains some interesting ideas about why priming of QS on a surface might be advantageous for *P. aeruginosa*, there is little direct mechanistic data on how *Lsr1* controls *lasR* expression or indeed how the pilin proteins control *lasR* expression in the presence of saturating 3-oxoC12-HSL.

We thank the reviewer for pointing out the intriguing aspects of our work and note that while we are also interested in the molecular mechanisms noted, those studies lie outside of our current paper's focus on the discovery that surface-associated cells are hypersensitive to QS signals.

Specific Points

1. The experimental set-up for the surface-associated vs planktonic uses a polystyrene petri dish. This raises some further questions. Is polystyrene a relevant surface for *P. aeruginosa* virulence/biofilms? Does the surface induction of lasR only occur on polystyrene? From the methods section, it is not clear how the surface-associated and planktonic cells were separated. Were the plates washed to remove weakly attached cells? Presumably there were far more planktonic cells than surface attached cells? Can the authors provide information on the numbers of surface associated and planktonic cells used for the subsequent transcriptome experiments? Do the cells have to be firmly attached to the surface to induce lasR expression or only transiently associated? Since the RNASeq/qPCR experiments only provide averages over the population, the use of a fluorescently labelled reporter gene fusion or flow cytometry would enable direct observation of the two cell populations to back up the transcriptome data and to determine whether the 'hypersensitive' expression of lasR was stochastic. Since transcriptional changes do not always manifest as translational changes, the paper could be strengthened by direct measurement of changes in LasR by Western blotting or ELISA type assays.

We thank the reviewer for raising these important questions, which we answer as follows:

- a. **In a previous study (Siryaporn et al., PNAS 2014) we showed that *P. aeruginosa* virulence is similarly induced by multiple surfaces independent of their surface chemistries, including a biological surface of a plant leaf. We have added this clarification to the text on line 133. As an additional test we also performed reporter assays on a different surface, glass, and found the same general trends (new Fig S2).**
- b. **We have clarified in the methods that the cells were washed to remove weakly-attached cells.**
- c. **For the transcriptome studies, we have included more details on our protocol. Importantly, all data were internally normalized to #reads/million, such that the absolute number of cells sequenced do not influence the relative abundances reported.**
- d. **We have used a fluorescent reporter of the target, *lasI*, and found that the responses are roughly normally distributed with no clear bimodality (Fig. S1), consistent with our model that surface sensing increases the general responsiveness of the cells to autoinducer.**
- e. **We have now generated a strain with a translational FLAG fusion to LasR and used Western blots to confirm that surface association also makes *P. aeruginosa* hypersensitive to QS with respect to LasR protein levels (Fig. S2).**

2. The authors use a number of *P. aeruginosa* mutants but none are genetically complemented to rule out secondary mutations. Previous work linking type IV twitching motility and quorum sensing was subsequently shown to be a consequence of compensatory mutations (Beatson et al 2002 J. Bact). Fig. 4C is of concern as the lasR and lasI deletion mutants show reduced twitching compared with the wild type.

We thank the reviewer for this point and have now shown complementation of the twitching defect for both LasR and LasI deletion mutants (Fig. S5).

3. The symbols in Fig. 2B (legend) are not explained. Fig. 2C (not 2D as in legend) is missing a 'no 3-oxoC12-HSL' control.

We thank the reviewer for catching this and have corrected it.

4. The term 'binding affinity' is used in Fig. S4 but it is not clear what this is referring to?

This is a good point, also raised by Reviewer 2. Since binding affinities are not a key part of our model or findings we have removed this unnecessarily confusing model figure.

REVIEWERS' COMMENTS:

Reviewer #1 (Remarks to the Author):

The revised paper by Chuang et al is much improved, and they have addressed my comments and suggestions. I have a few very small issues.

1. Regarding the twitching, the authors say in line 277 that all the mutants maintained their predicted twitching phenotypes. I was unclear on what were the expected predictions. For the pilW knockout that is surface locked (in terms of quorum sensing), it wasn't obvious to me that it should have reduced twitching. Does this mean that twitching and QS are two completely separate pathways?
2. Figure 3 was still difficult to read on a printed page. The axis and graph legends were quite small.

Otherwise I believe the manuscript is ready for publication.

Reviewer #3 (Remarks to the Author):

The authors have strengthened their interesting manuscript with additional experiments, controls, methodological information and improved presentation. There are a two minor issues remaining:

1. The authors have still not provided convincing evidence that surfaces 'hypersensitize' *P. aeruginosa* to quorum sensing. 'Sensitize' or preferably as the authors suggest on lines 199-200, 'surface-primed QS' is a much more accurate term to denote the phenomenon described.
2. In the authors rebuttal point a, they state that they now provide clarification with respect to different surfaces on line 133 and a new figure (Fig. S2) where reporter assays are performed on glass. In the revised version, there is no additional relevant text on line 133 nor does Fig. S2 show a reporter assay on glass. Point b – the method by which the cells are recovered from the surface, harvested and washed is extremely important. It has now been added to the revised Methods but is a bit lost in the section on Western blot analysis. It ought to be placed in upfront in the methods as it applies to the other assays used.

REVIEWERS' COMMENTS:

Reviewer #1 (Remarks to the Author):

The revised paper by Chuang et al is much improved, and they have addressed my comments and suggestions. I have a few very small issues.

1. Regarding the twitching, the authors say in line 277 that all the mutants maintained their predicted twitching phenotypes. I was unclear on what were the expected predictions. For the *pilW* knockout that is surface locked (in terms of quorum sensing), it wasn't obvious to me that it should have reduced twitching. Does this mean that twitching and QS are two completely separate pathways?

Yes, *pilW* has low twitching that is similar to the loss of the other *pil* mutants (as has been previously reported and confirmed here in Fig. S4). Indeed this suggests that the effects of loss of *pilW* on QS and twitching are opposite. Whether this reflects these pathways being completely separate, as the reviewer suggests, or a function of a more complicated feedback system remains unclear and beyond the scope of the current work, so we have not elaborated on this interesting open question.

2. Figure 3 was still difficult to read on a printed page. The axis and graph legends were quite small.

Size was increased

Otherwise I believe the manuscript is ready for publication.

Reviewer #3 (Remarks to the Author):

The authors have strengthened their interesting manuscript with additional experiments, controls, methodological information and improved presentation. There are a two minor issues remaining:

1. The authors have still not provided convincing evidence that surfaces 'hypersensitize' *P. aeruginosa* to quorum sensing. 'Sensitize' or preferably as the authors suggest on lines 199-200, 'surface-primed QS' is a much more accurate term to denote the phenomenon described.

The reviewer makes a good point and we have removed the term "hypersensitive" from the paper, replacing it with "increased sensitivity" or "surface-primed QS" where appropriate.

2. In the authors rebuttal point a, they state that they now provide clarification with respect to different surfaces on line 133 and a new figure (Fig. S2) where reporter

assays are performed on glass. In the revised version, there is no additional relevant text on line 133 nor does Fig. S2 show a reporter assay on glass. Point b – the method by which the cells are recovered from the surface, harvested and washed is extremely important. It has now been added to the revised Methods but is a bit lost in the section on Western blot analysis. It ought to be placed in upfront in the methods as it applies to the other assays used.

We apologize for this oversight. We have now added a sentence starting on line 187: “These fluorescent reporter assays were performed using a glass surface, while the RNA-seq and qRT-PCR assays were performed using polystyrene surfaces, indicating that the sensitization of QS is not dependent on the composition of the surface.” We have also addressed point “b” by providing a more detailed description of the isolation of planktonic and surface-attached cells for RNA-seq and qRT-PCR experiments at the beginning of the Methods section.